# Converging deep learning and human-observed tumor-adipocyte interaction as a biomarker in colorectal cancer
Nic G. Reitsam [1,2,3] ✉, Bianca Grosser[1,2], David F. Steiner [4], Veselin Grozdanov[5], Ellery Wulczyn[3], Vincenzo L'Imperio [6], Markus Plass [7], Heimo Müller [7], Kurt Zatloukal [7], Hannah S. Muti[3,8], Jakob N. Kather [3,9,10,11] & Bruno Märkl [1,2]

## Abstract

**Background** Tumor-Adipose-Feature (TAF) as well as SARIFA (Stroma AReactive Invasion Front Areas) are two histologic features/biomarkers linking tumor-associated adipocytes to poor outcomes in colorectal cancer (CRC) patients. Whereas TAF was identified by deep learning (DL) algorithms, SARIFA was established as a human-observed histopathologic biomarker.
**Methods** To study the overlap between TAF and SARIFA, we performed a systematic pathological review of TAF based on all published image tiles. Additionally, we analyzed the presence/absence of TAF in SARIFA-negative CRC cases to elucidate the biologic and prognostic role of a direct tumor-adipocyte contact. TCGA-CRC gene expression data is investigated to assess the association of *FABP4* (fatty-acid binding protein 4) and *CD36* (fatty-acid translocase) with both TAF and CRC prognosis.
**Results** By investigating the TAF/SARIFA overlap, we show that many TAF patches correspond to the recently described SARIFA-phenomenon. Even though there is a pronounced morphological and biological overlap, there are differences in the concepts. The presence of TAF in SARIFA-negative CRCs is not associated with poor outcomes in this cohort, potentially highlighting the importance of a direct tumor-adipocyte interaction. Upregulation of *FABP4* and *CD36* gene expression seem both linked to a poor prognosis in CRC.
**Conclusions** By proving the substantial overlap between human-observed SARIFA and DL-based TAF as morphologic biomarkers, we demonstrate that linking DL-based image features to independently developed histopathologic biomarkers is a promising tool in the identification of clinically and biologically meaningful biomarkers. Adipocyte-tumor-cell interactions seem to be crucial in CRC, which should be considered as biomarkers for further investigations.

## Plain language summary

Different methods exist in assessing samples removed from cancer patients during surgery. We linked two independently established tissue-based methods for determining the outcome of colorectal cancer patients together: tumor adipose feature (TAF) and Stroma AReactive Invasion Front Areas (SARIFA). SARIFA as biological feature was observed solely by humans and TAF was identified by the help of a computer algorithm. We examined TAF in many cancer slides and looked at whether they showed similarities to SARIFA. TAF often matched SARIFA, but not always. Interestingly, these methods could be used to predict outcomes for patients and are associated with specific gene expression involved in tumor and fat cell interaction. Our study shows that combining computer algorithms with human expertize in evaluating tissue samples can identify meaningful features in patient samples, which may help to predict the best treatment options.

With a broad spectrum of clinical courses based on the complex interplay of genetic, molecular, and microenvironmental factors, the adequate and tailored treatment strategy for colorectal cancer (CRC) patients remains a formidable challenge in modern oncology[1–3]. Currently, patient stratification is mainly based on the American Joint Committee on Cancer/Union for International Cancer Control/Tumor Node Metastasis (AJCC/UICC/TNM) classification, supported by histopathological features such as grading or tumor budding. On top of that, molecular biomarkers, focusing on the specific biology of the tumor cells itself, such as *KRAS* or *BRAF* mutational status, mismatch repair (MMR), or microsatellite status have found their way into the clinic. Yet, all these routinely used biomarkers are still not enough to sufficiently guide treatment decisions[4]. Therefore, gene expression-based approaches such as consensus molecular subtyping (CMS) have been established[5]. In this context, it is important to highlight

---

that these molecular and transcriptomic approaches are based on challenging assays, that go far beyond on for every cancer patient's routinely available hematoxylin & eosin (H&E) histology.

Still, these molecular and transcriptomic approaches could reveal that CRCs, like other malignancies, harbor a rich microenvironment consisting of diverse stromal cells, that actively shape disease progression as well as therapeutic response[6]. Among these stromal constituents, adipocytes have emerged as unexpected actor in CRC[7]. However, the intricate crosstalk between tumor cells, adipocytes, and the role of lipid metabolism have been extensively studied in the past—mainly with a focus on basic functional mechanisms or the association between cancer and obesity[8–10]. On H&E histopathology, the direct contact between adipocytes and tumor cells can be easily studied, and therefore, we defined these areas as so-called Stroma AReactive Invasion Front Areas (SARIFA), where there is no inflammatory infiltrate or desmoplastic stroma reaction preventing the formation of a direct tumor-adipocyte interaction at the prognostically important invasive tumor margin[11–14]. Indeed, we could prove that the presence of SARIFA is not only associated with a poor outcome in colorectal as well as gastric cancer but also with an upregulation of fatty-acid metabolism[11,12,14,15]. Interestingly, SARIFA-positive gastric cancers as well as CRCs seem to show an altered immune response[12,14,16].

In recent years, the field of biomarker development has undergone a transformative shift, driven by the advances in deep learning (DL) and the aforementioned growing understanding of the tumor-stroma interactions. Many DL models, which are mostly based on routinely available H&E slides, could impressively not only predict patient outcome in CRC[17–20] but also predict molecular features, such as MSI status[21–23] or imCMS subtypes[24]. Even though these DL algorithms are very potent in predicting prognosis and molecular features, due to their partly 'blackbox' nature and the need for a computing-heavy infrastructure their implementation into the clinical diagnostic process is still very limited. However, by resolving molecular predictions, such as imCMS[24], on a spatial tile level, interesting insights into underlying tumor biology, including tumor heterogeneity, as well as increased interpretability highlight the great potential of DL models to gain biological insights. Beyond this, imCMS for example could potentially help to identify patients who are most likely to benefit from neoadjuvant therapy[25], further demonstrating the great potential of DL algorithms in CRC pathology.

Interestingly, our SARIFA research approach commenced with this large-scale application of DL algorithms to histopathologic CRC slides. Serendipitously, one DL-based approach simultaneously led to the identification of a so-called tumor adipose feature (TAF)[19], similar to what we define as SARIFA.

Therefore, we present here the convergence of these two distinct yet complementary concepts: the discovery of tumor adipose feature (TAF) through DL algorithms and the identification of Stroma AReactive Invasion Front Areas (SARIFA) as a human-observed histopathologic morphological biomarker.

The current study aims at investigating the overlap between TAF and SARIFA by a detailed histopathologic review and further analysis based on gene expression and thereby provides an example of how histopathology and DL can reinforce each other in the development of histopathologic H&E-based biomarkers.

## Methods
### Ethics Statement
The experiments in this study are in compliance with the Declaration of Helsinki and the International Ethical Guidelines for Biomedical Research Involving Human Subjects by the Council for International Organizations of Medical Sciences (CIOMS). The patient sample collection in each cohort was separately approved by the respective institutional ethics boards. As in our previous publication[11], the overall analysis in this study has been approved by the Ethics Board at the Medical Faculty of Technical University Dresden (BO-EK-444102022). The original studies from which patient samples were derived obtained informed consent from the participants.

### Data collection & availability
The aim of this study was to investigate the overlap between human-observed SARIFA and DL-based TAF as histopathologic biomarkers in CRC pathology. TAF has been recently described as prognostic image feature in colorectal cancer, that could be identified by using DL algorithms[19] and was also validated by human pathologists[26]; an additional DL-based study could link a very similar image feature to lymph node metastasis in CRC[27]. All source material is fully available in the respective publications[19,26,27] and in the Supplementary Material and Supplementary Data 1–3 included in this manuscript. To assess the overlap between SARIFA and TAF, several TAF patches, that were published with the corresponding articles (either in the main manuscript or as supplementary material), were investigated[19,26]. Detailed information on the investigated image patches and WSIs can be found in the next section (Assessment of TAF/SARIFA Overlap). Clinical and molecular data as well as image data of TCGA COAD (colonic adenocarcinoma) and READ (rectal adenocarcinoma) is publicly available at https://www.cbioportal.org/[28–30], and it is permitted to use this for further analyses and studies (https://docs.cbioportal.org/user-guide/). All data that has been generated during this study is included in this article and/or available on request from the corresponding author (NGR).

### Assessment of TAF/SARIFA overlap
In the present article, we assess whether TAF corresponds to what we previously defined as SARIFA. SARIFA-positivity is defined as the direct contact between tumor cells (at least one single tumor gland or more than five tumor cells) and adipocytes without intervening inflammatory infiltrate or desmoplasia (SARIFA-positive). If there is no direct tumor-adipocyte contact (at the invasion front), the case is classified as SARIFA-negative. TAF is a biomarker identified by DL and can be described as the proximity of CRC tumor cells to adipose tissue[19].

Several TAF patches were reviewed for the current study. Out of the initial publication by Wulczyn et al.[19], 175 TAF patches were assessed: Supplement 1 (closest to centroids, $n = 25$; randomly sampled TAF patches, $n = 25$), Supplement 2 (practice patches, $n = 25$), Supplement 3 (assessment patches, $n = 100$). We also investigated all non-TAF patches (Supplement 2, practice patches, $n = 25$; Supplement 3, assessment patches, $n = 100$). With regards to the pathologist validation study[26], we examined 3 examples of TAF (Fig. 1), that even provided spatial information with an image of the whole slide; and also the supplementary online content ($n = 7$). In the study by Krogue et al.[27], 35 sample patches with an image feature, similar to TAF, were observed (Fig. 2: temporal validation, $n = 5$; Fig. 2: external validation 1a, $n = 5$; Supplementary Fig. S2, $n = 25$).

Patch sizes are as follows: Wulczyn et al.[19] $256 \times 256$ pixels at 5× magnification, Krogue et al.[27] $289 \times 289$ pixels obtained at 10×, L'Imperio et al.[26] different resolution and size but scale bars provided in the original publication for transparency.

Moreover, we assessed all SARIFA-negative CRCs ($n = 138$[11]) in TCGA-COAD and TCGA-READ[28,31] for the presence of TAF. Given that essentially all SARIFA patches are also TAF patches based on the definitions and observations of these features, we classified SARIFA-positive cases as TAF-present. SARIFA status in TCGA-CRC was established as described in a previous study[11]. Pathological review was performed by NGR and BG. Interobserver agreement reached a very good kappa score of 0.87 (Table S1). In cases of discrepancy, a consensus was established. The most common reason for discrepancy was the question whether there is a direct tumor-adipocyte contact or a small rim of stroma in between (few) tumor cells and adipocytes. The assessment was supervised by all board-certified pathologists among the authors (DS, VL, KZ, BM). The study design is visualized in Fig. 1.

### mRNA expression analysis & CMS/PDS/IS subtyping
To examine the relevance of lipid metabolism in CRC, expression of *FABP4* (fatty-acid binding protein 4) and *CD36* (FAT, fatty-acid translocase) in CRCs was analyzed for association with the presence or absence of TAF as

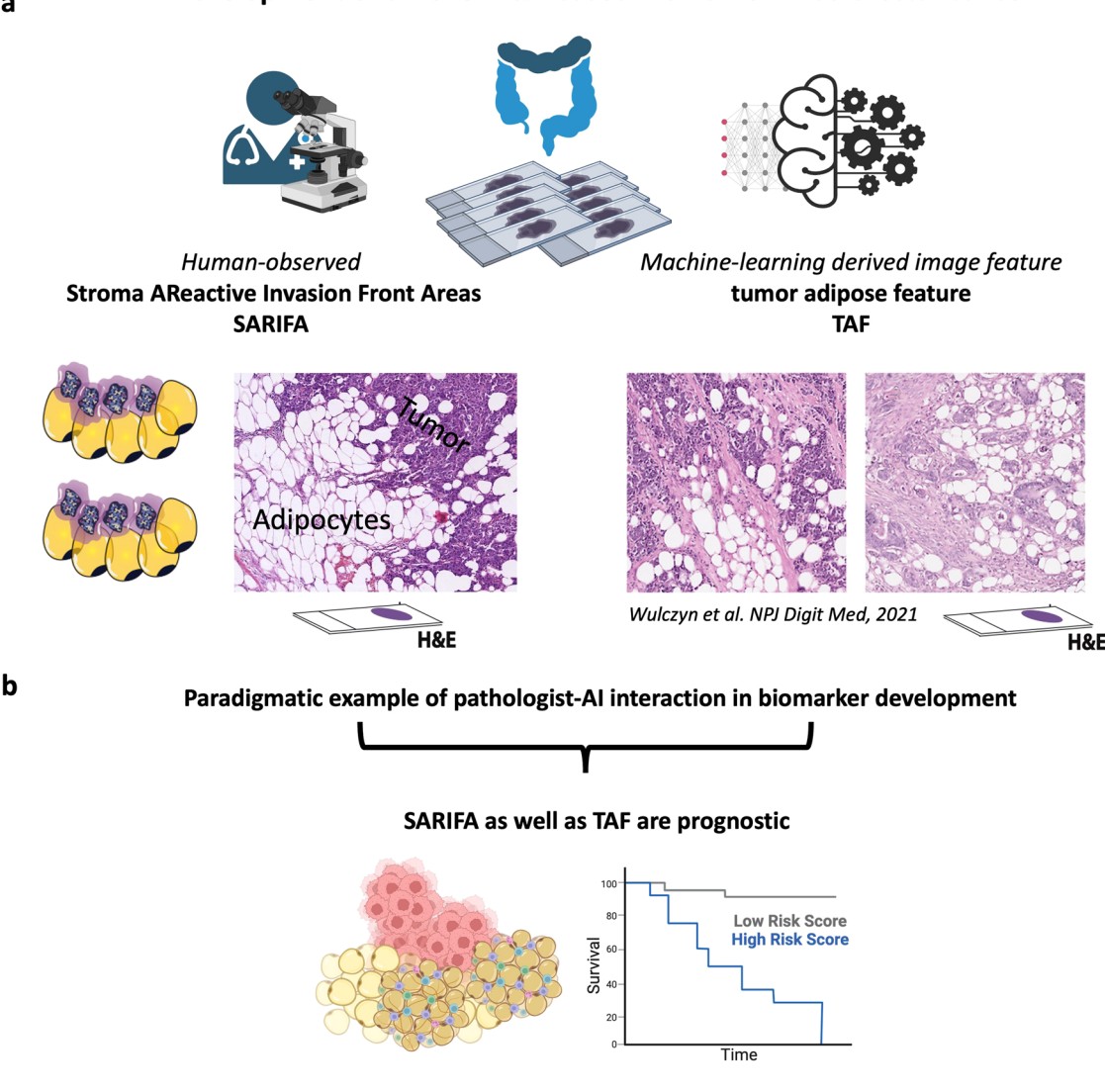

**Fig. 1 | TAF and SARIFA as prime examples of pathologist-AI interaction in biomarker development. a** SARIFA is a human-observed biomarker based on H&E histopathology and defined by the direct contact between tumor cells and adipocytes at the invasion front. Independently of SARIFA, deep learning algorithms trained to predict survival from H&E slides helped to uncover TAF, an image feature characterized by the proximity of tumor cells and adipocytes, as a similarly important morphological feature in colorectal cancer. **b** Both SARIFA as well as TAF are prognostic histology-based biomarkers in colorectal cancer and can potentially guide treatment decisions by identifying high-risk patients. **c** In this study, we aimed to characterize the TAF/SARIFA overlap by a thorough pathological review and also performed further analyses to study the role of direct-tumor adipocyte interaction with regards to tumor biology, and in particular *FABP4* and *CD36* expression. AI artificial intelligence, CRC colorectal cancer, H&E hematoxylin & eosin, SARIFA Stroma AReactive Invasion Front Area, TAF tumor adipose feature examples of TAFs were previously published by Wulczyn et al.[19] under Creative Commons Attribution 4.0 International License. *Created with BioRender.com*

well as with prognosis. *FABP4* as well as *CD36* are strongly associated with SARIFA positivity[11,12,14] and play an important role in lipid metabolism[8,9]. For gene expression analysis, RNA expression z-scores relative to normal samples (log RNA Seq V2 RSEM) as well as batch-normalized RNA-seq data from Illumina HiSeq_RNASeq_V2, along with overall survival (OS), disease-specific survival (DSS) and progression-free survival (PFS) of the corresponding patients as endpoints, were accessed from TCGA cohorts COAD and READ via cBioPortal[29,30]. Two cut-offs for stratification of TCGA-CRC patients based on gene expression were applied: top tertile (1/3) of expression, and a stricter 10% cut-off to further evaluate for a possible expression-dependent effect on prognosis (e.g., dose-dependency as potential biological mechanism). As these analyses are only aimed at providing insights into the possible biological implications of SARIFA/TAF, and not for diagnostic use itself, no optimized cut-offs were determined.

Additionally, differential gene expression analysis for TCGA-CRC samples ($n = 196$, refer also to our previous publication[11]) was performed

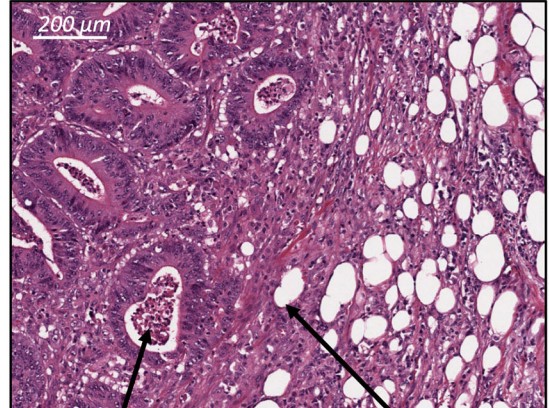

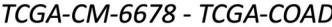

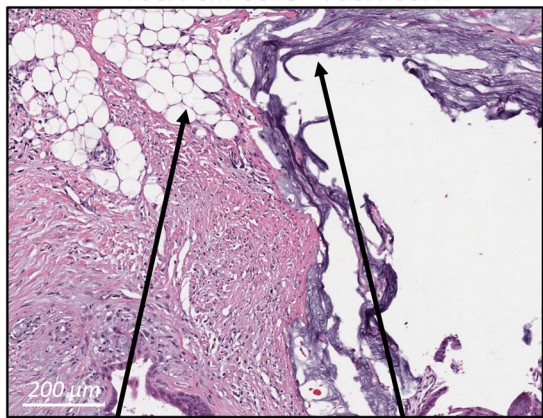

**Fig. 2 | Important role of direct tumor-adipocyte interaction. a** As TAF is characterized by the proximity of tumor cells and adipocytes, but not necessarily the direct contact between tumor cells and adipocytes, there were also SARIFA-negative CRCs displaying TAF (32.6%) in our pathologic review. **b** Presence of TAF was not significantly associated with poorer outcomes in SARIFA-negative CRC in TCGA cohorts COAD and READ (each $p > 0.05$), which potentially indicates the important

role of a direct tumor-adipocyte interaction. Upper panel: overall survival, middle panel: disease-specific survival, lower panel: progression-free survival. For each microscopic image appropriate scale bars are provided; scale bars indicate 200 μm. COAD colonic adenocarcinoma, CRC colorectal cancer, NOS not otherwise specified, TAF tumor adipose feature, TCGA The-Cancer-Genome-Atlas, SARIFA Stroma AReactive Invasion Front Area, READ: rectal adenocarcinoma.

with DESeq2[32] with counts rounded to an integer and Wald test without covariates (for different designs: 1. TAF presence vs absence, 2. SARIFA-positive vs SARIFA-negative, and 3. a combined score [SARIFA-positive/TAF-present = double positive vs SARIFA-negative/TAF-present = only TAF vs SARIFA-negative/TAF-absent = double negative]). Functional gene set enrichment was performed with ShinyGo 0.80[33] (http://bioinformatics.sdstate.edu/go/) with an FDR (false-discovery cut-off) of 0.05 (Curated.Reactome).

We further aimed to link gene expression-based established CRC consensus molecular subtypes (CMS)[5], pan-cancer immune subtypes (IS)[34], as well as just recently published pathway-derived subtypes (PDS)[35,36] to SARIFA/TAF as H&E biomarkers. CMS calls with regards to SARIFA-status were previously published[11]. Subtyping was performed on normalized gene-expression counts based on our previous publication[11] (number of

CRC cases: $n = 196$) by deploying R packages CMScaller[37], PDSclassifier[36], and ImmuneSubtypeClassifier[34].

## Statistical analysis

Proportionate frequencies are given as percentages (%). Estimates of Kaplan-Meier survival rates were compared using pairwise log-rank tests. Continuous variables were compared using the Wilcoxon rank-sum test (two-sided). $p$-values < 0.05 were considered as statistically significant. For mRNA expression analysis $q$-values are reported to incorporate multiple testing corrections (using a false discovery rate detection approach). Statistical analysis was performed by using R (v4.2.2; R Foundation for Statistical Computing, Vienna, Austria). Additional R packages used were survminer, survival, tidyverse, ggplot2.

## Table 1 | SARIFA/TAF overlap: pathologic review

| References[19,26,27] | Category | Overlap: SARIFA/TAF (in %) | Reasons for discrepancy |
|---|---|---|---|
| 1) Wulczyn et al.[19] | Supplement 1: closest to centroids | 25/25 (100%) | 1. intervening stroma or inflammation between tumor cells and adipocytes |
| | Supplement 1: randomly sampled TAF patches | 16/25 (64%) | 2. mucinous tumors where no vital tumor cells were directly adjacent to adipocytes but only mucin |
| | Supplement 2: practice patches | 13/25 (53%) | 3. inflammatory infiltrate or peripheral ganglia in adipose tissue that mimic cancer cells (this may be also associated with poor prognosis, at least shown for inflamed adipose tissue[43]) |
| | Supplement 3: assessment patches | 63/100 (63%) | Also it must be kept in mind that TAF are random tiles and do not necessarily depict the invasion front; however, the tumor-fat interface is often the invasion front. |
| 2) L'Imperio et al.[26] | Fig. 1: Examples of TAF | 3/3 (100%) | |
| | Supplementary online content | 6/7 (86%) | |
| 3) Krogue et al.[27] | Fig. 2: Temporal validation | 0/5 (0%) | |
| | Fig. 2: External validation 1a | 3/5 (60%) | |
| | Supplementary Fig. S2 | 8/25 (32%) | |

Note: In 3), a similar feature to TAF was identified and described as *"predominantly adipose and inflammatory cells with occasional tumor cells."* Accordingly, the overlap between SARIFA and this feature was assessed even though this feature did not necessarily match the exact TAF classification as in 1).
Overall, our findings indicate a relevant histological overlap between SARIFA and TAF—while there are also some subtle differences.

### Reporting summary

Further information on research design is available in the Nature Portfolio Reporting Summary linked to this article.

## Results

### TAF and SARIFA show a substantial histological overlap

To assess the exact overlap between SARIFA as human-derived histo-pathologic biomarker and TAF as DL-based image feature in colorectal cancer specimens (Table 1), we visually reviewed 175 TAF patches from the initial study discovering TAF[19], which aimed at predicting survival from CRC whole slide images (WSI). In the TAF patches closest to centroids, which are most representative of their cluster, e.g., of TAF, there was a 100% overlap; in all of these patches, at least 5 tumor cells or at least one tumor gland were directly adjacent to adipocytes, which is consistent with our SARIFA definition. In the other image patches, the TAF/SARIFA overlap frequency ranged from 53% (Supplement 2: practice patches) to 64% (Supplement 1: randomly sampled TAF patches), which shows that the TAF does not necessarily depict SARIFA. We also qualitatively assessed the reasons for this discrepancy and could observe that most TAF patches are, consistent with their nomenclature, indeed showing tumor cells close to adipocytes – however, there is often some intervening stroma or inflammation in between tumor cells and adipocytes, which is not in line with our precise SARIFA definition, that demands for a direct tumor-adipocyte interaction. Additionally, in CRCs with mucinous components at the tumor invasion front often only the mucin itself but no vital tumor glands are directly adjacent to adipocytes, which diverges from our SARIFA definition. Furthermore, in some cases classified as TAF-present, inflammatory infiltrate or peripheral ganglia in adipose tissue that mimics cancer cells are depicted, also not fulfilling the criteria of SARIFA-positivity. Overall, 66.9% of all TAF patches in the initial discovery study showed SARIFA, highlighting a relevant overlap. In non-TAF patches ($n = 25$ practice patches & $n = 100$ assessment patches), this similarity is even more pronounced, as there are no TAF-negative but SARIFA-positive patches (SARIFA-positivity in non-TAF patches: 0% [0/125]).

Based on the TAF discovery study, subsequent work demonstrated reliability and prognostic impact of TAF recognition by pathologists, showing the translatability of DL-based histological features in the routine practice (human-in-the-loop)[26]. Here, the representative TAF cases (see: Fig. 1: Examples of TAF) all are in line with our SARIFA definition (3/3, 100% overlap). In the supplementary material, further TAF patch examples guiding the pathologists in their learning process are shown. Out of these TAF patches, 6/7 (86%) depict SARIFA. Interestingly, compared to the initial biomarker discovery study[19], where TAF was reported on random tiles without spatial resolution, the stemming pathologist-based work showed a prevalence of TAF at the invasion front[26], which is in most (but not all) cases the equivalent of the tumor-fat interface.

In a consecutive study, Krogue et al. identified that a similar feature to TAF, which was described as *"predominantly adipose and inflammatory cells with occasional tumor cells"* seems to be relevant for predicting lymph node metastasis from primary colon cancer H&E slides[27]. As SARIFA-positivity is similarly associated with a higher frequency of lymph node metastases[11,13], the overlap between SARIFA and this feature was assessed. Here, overall 35 patches could be assessed, of which 11 showed our SARIFA phenomenon (31%).

A more detailed overview of the discrepancies in the pathologist validation study and the similar image feature identified by Krogue et al[27] can be found in Table S2.

Comparing the frequency of colon cancers displaying TAF (46%[26]) and SARIFA-positivity (25%[13]), our impression is strengthened that there is an overlap between TAF and SARIFA—however, as TAF is defined by the sole proximity between tumor cells and adipocytes and not the direct contact, the TAF frequency is higher than the SARIFA frequency.

### Direct tumor-adipocyte interaction seems to be prognostically relevant

Based on these findings, we wanted to investigate the role of the presence of TAF in SARIFA-negative CRC cases. Therefore, we visually reviewed all SARIFA-negative CRCs within TCGA cohorts COAD and READ ($n = 138$[11]) for the presence of TAF. Here, 45 (32.6%) of all SARIFA-negative cases presented with TAF, which leads to an overall TAF frequency of 55% in TCGA-CRC as all SARIFA-positive CRCs ($n = 69$ in TCGA-CRC) logically also display TAF. However, here it must be noticed that TAF in SARIFA-negative cases are defined by tumor cells in close proximity to adipocytes without direct tumor-adipocyte interaction (Fig. 2a), which is not always easy to evaluate, as there is no clear cut-off for how close tumor cells and adipocytes have to be. In Kaplan-Meier analysis no significant differences regarding OS, DSS, and PFS in the subgroup of SARIFA-negative CRCs stratified by the absence/presence of TAF could be observed (each $p > 0.05$, Fig. 2b), suggesting that most of the survival differences are potentially driven by the direct tumor-adipocyte interaction. However, with regards to OS, at least a trend ($p = 0.22$) for a shorter survival of SARIFA-negative CRCs with presence of TAF can be seen.

### Underlying biology of SARIFA/TAF & their overlap in CRC

TAF and SARIFA as morphologic biomarkers are likely to display a certain tumor biology (potentially linked to lipid metabolism), that may be causally relevant for the observed changes in prognosis.

Even though consensus molecular subtype (CMS) 3 is considered the metabolic CRC subtype with an upregulation of several metabolic pathways (not only fatty-acid metabolism but also glucose, fructose, nitrogen,… metabolism) and activating KRAS mutations[5], we have already demonstrated that SARIFA-positive CRCs show an enrichment for CMS1/4 (immune/mesenchymal) and not for CMS3[11], indicating that SARIFA-positivity is not only associated with changes in fatty-acid metabolism[11,12] but also dysregulated immunity[14,16] and stromal changes[11]. By investigating associations between CMS-calls and TAF ($n = 196$ CRCs, TCGA-CRC), we also could, as expected, observe similar TAF-dependent differences in CMS-calls, with an enrichment of CMS1/4 (immune/mesenchymal, $p = 0.022$, Fig. 3C). By establishing pathway-derived subtypes (PDS)[36], we again observed statistically significant SARIFA/TAF-dependent differences (TAF: $p = 0.0157$, SARIFA: $p = 0.004747$), with an enrichment of PDS2 among SARIFA-positive CRCs and CRCs with TAF (Fig. 3d, Supplementary Fig. 1a), highlighting again the stromal & immune relevant changes[35,36] associated with histologically visible tumor-adipocyte interaction.

Interestingly, compared to CMS/PDS, pan-cancer immune subtyping (IS)[34] did not differ significantly based on SARIFA-status or TAF (each $p > 0.5$), with most CRC cases in general being of "*wound healing*" or "*interferon-gamma dominant*" subtype (Supplementary Fig. 1b), potentially indicating that site-specific transcriptomic CRC classifiers (CMS & PDS) may better correlate with prognostically and biologically relevant histologic features (SARIFA/TAF).

As we could previously show an upregulation of *CD36* and *FABP4* on gene as well as protein level, both closely linked to lipid metabolism, in SARIFA-positive gastric as well as colorectal cancers[11,12,14], we wanted to investigate *CD36* and *FABP4* expression in CRCs with and without TAF in TCGA-CRC based on our pathologic review. Here, we could observe a significant upregulation of *CD36* and *FABP4* mRNA expression based on z-scores relative to normal samples (both $p < 0.05$) in CRCs with TAF compared to those without TAF (Fig. 3a, b).

To further investigate the biological profiles of SARIFA/TAF, we performed differential gene expression analysis with the transcriptomic profiles of 196 TCGA-CRC samples with TAF-status, which revealed a broad dysregulation of gene expression (TAF presence vs absence: 1483 genes upregulated, 755 genes down-regulated, $q < 0.05$, no LFC threshold; LFC: log fold change, see Supplementary Data 2)—quite similar to what previously observed with regards to SARIFA-status[11]. The significant upregulation of both *FABP4* and *CD36* in CRCs with TAF could be confirmed (*FABP4*: LFC 2.19, $q = 6.65\text{E-}08$; *CD36*: LFC 1.93, $q = 0.0001$).

We further investigated the overlap of significantly upregulated genes (LFC > 1, $q \leq 0.05$) between the differential gene expression analysis based on TAF-status and based on SARIFA-status. The overlap frequency was 32%. Next, we checked for all genes that are upregulated in SARIFA-positive but not in CRCs with TAF, and vice versa (LFC > 1, $q \leq 0.05$ again as criteria). The mutually exclusive upregulated genes are listed in Supplementary Data 1. When performing pathway analysis (Curated.Reactome via ShinyGo 8.0[33]) with these mutually exclusive upregulated genes, no significant enrichment for the differing genes associated with TAF presence could be found (FDR cutt-off 0.5), indicating that they do not belong to a specific biological process. When investigating the only in SARIFA-positive upregulated genes, several pathways were significantly enriched; out of those, several pathways belong to lipid signaling, indicating a more pronounced upregulation of lipid metabolism in SARIFA-positive CRCs compared to CRCs with TAF (Fig. 3d). The overlapping upregulated genes showed significant association to several biological pathways, again highlighting the close biological overlap between SARIFA and TAF, mainly with regards to lipid metabolism but also immune/stroma relevant pathways (Fig. 3c).

Furthermore, we created a combined score (SARIFA-positive/TAF-present = double positive, SARIFA-positive/TAF-present = only TAF,

SARIFA-negative/TAF-absent = double negative), and compared *FABP4* and *CD36* expression among these three groups (Supplementary Fig. 2). As expected, there was a statistically significant increase in *FABP4* and *CD36* expression between SARIFA-positive/TAF-present and SARIFA-negative/TAF-absent CRCs (*FABP4*: $p = 0.0088$, *CD36*: $p = 0.0029$), and we could observe a numeric trend-wise increase from SARIFA-negative/TAF-absent over only TAF-present to SARIFA-positive/TAF-present CRCs (p-values all above 0.05; for SARIFA-positive/TAF-present vs only TAF *FABP4*: $p = 0.16$), with only TAF-present CRCs showing intermediate expression levels.

Differential gene expression analysis between TAF-present/SARIFA-negative CRCs (only TAF) and SARIFA-positive (and TAF-present, double positive) CRCs revealed among the 60 upregulated genes in the double positive CRCs (LFC > 0, no LFC threshold, $q \leq 0.05$) several genes that are involved in lipid metabolism (*FABP4, PLIN1, ADIPOQ, PLIN4, CD36*; see Supplementary Table 3, see Supplementary Data 2), again suggesting that direct tumor-adipocyte interaction may not only have prognostic but also biologic implications.

Based on these findings, we wanted to investigate the prognostic value of *CD36* and *FABP4* expression regardless of SARIFA-status and the presence/absence of TAF. Similarly to the presence of TAF and SARIFA-positivity, upregulation of *CD36* and *FABP4* is associated with worse outcomes in univariate Kaplan-Meier analysis in TCGA-CRC regarding most endpoints (Fig. 4), strengthening the association between an increased lipid metabolism and an aggressive tumor behavior.

In line with our previous studies on the prognostic value of SARIFA/TAF[11,13,19,26], we could validate this negative prognostic value of SARIFA-positivity/presence of TAF now on the sample subset ($n = 196$[11]), which we used for differential gene expression analysis (each $p < 0.05$, log-rank test, refer to Supplementary Fig. 3). Subsequently stratifying for different subgroups (SARIFA-positive, SARIFA-negative, TAF-present, TAF-absent) did not reveal further stratification based on *CD36* and *FABP4* expression, particularly as certain subgroups became smaller (refer to Supplementary Fig. 4 for Kaplan–Meier curves for CRCs with TAF based on *CD36/FABP4* expression; p-values of log-rank tests can be found in Supplementary Table 4). These findings suggest that the prognostic impact of *CD36* and *FABP4* may partly be encompassed within the SARIFA/TAF classifications; however, also other pathways (beyond lipid metabolism) seem here relevant (Fig. 3c, d).

## Discussion

In this study, we link, as far as we know for the first time, two independently established biomarker approaches in CRC pathology: DL-derived tumor adipose feature (TAF) and human-observed Stroma AReactive Invasion Front Areas (SARIFA). Even though both concepts arose independently, they share striking similarities, and both highlight the so far under-appreciated important prognostic role of tumor-adipocyte interaction in CRC. SARIFA is characterized by the direct interaction between tumor cells and fat cells without intervening stroma, is prognostic in CRC, and shows the advantage of an easy, fast, and reliable H&E-based assessment without producing further costs[11,13]. Additional studies not only validated its prognostic role in gastric cancer but also closely linked SARIFA to a distinct tumor biology relying on lipid metabolism and immune alterations[11,12,14,16]. Simultaneously, TAF was discovered through a DL model trained on H&E slides to predict survival from CRC slides[19]. Subsequently, TAF was validated as a prognostic image feature that can be learned and scored by pathologists[26]. The present article provides a detailed comparison of SARIFA and TAF (Table 2). It should be noted that we have first evidence that SARIFA may be relevant also in other tumor entities, like pancreatic or prostate adenocarcinoma[38,39]. Interestingly, similar DL-based approaches also support the important prognostic role of tumor-adipocyte interaction, not only on survival (see Figure 5G in ref. 18), but also on lymph node metastasis in colorectal[27] as well as gastric cancer[40]. Similarly, Jiang et al. highlighted the importance of tumor-infiltrated fat for a poor prognosis in their end-to-end DL prognostication model (see Jiang et al. Supplementary

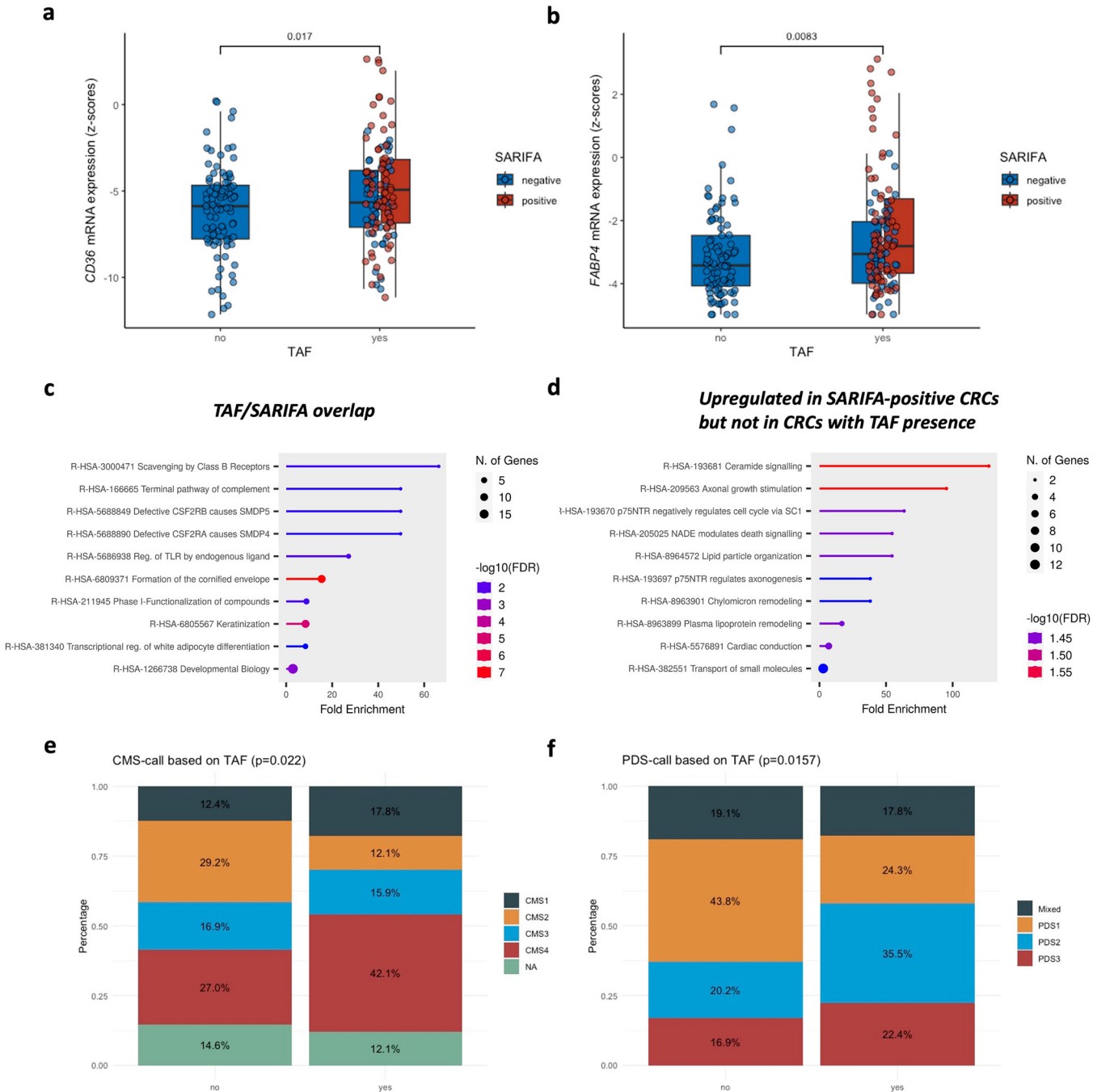

**Fig. 3 | Overlap of SARIFA and TAF leads to a similar biologic profile. a, b** *CD36* and F*ABP4* mRNA Expression in TCGA-CRC stratified by the presence/absence of TAF. Similarly to what we have already observed in SARIFA-positive CRCs[10], the presence of TAF, which of course shows substantial overlap with SARIFA-positivity (*n* = 69), is associated with higher *CD36* (**a** *p* = 0.017) and *FABP4* (**b** *p* = 0.0083) mRNA expression based on z-scores relative to normal samples. 45 SARIFA-negative CRCs (*n* = 138) showed TAF based on our pathologic review of TCGA-CRC (examples are displayed in Fig. 2a). For the additional analysis regarding *FABP4* and *CD36* expression with regards to our combined SARIFA/TAF score, please refer to Supplementary Fig. 2. **c, d** Pathway analysis via *ShinyGo*[33]. *C*. Pathway analysis of genes that were upregulated in both SARIFA-positive CRCs and CRCs with TAF presence (overlapping upregulated genes). *D*. Pathway analysis of only in SARIFA-positive CRCs upregulated genes (and not in CRCs with TAF presence, mutually exclusive). Genes that were only upregulated in presence of TAF did not show significant association to any biological pathway. Differential gene expression

analysis was performed via *DESeq2*[32] (SARIFA-positive vs SARIFA-negative and TAF presence vs TAF absence). Significantly upregulated genes (LFC > 1, *q* ≤ 0.05) between the analyses were then compared (overlapping vs differing [Supplementary Data 1]), and pathway analysis (Curated.Reactome) was performed. **e, f** CMS[37]- and PDS[36]-calls (n = 196 CRCs) show significant differences (*p* = 0.022 and *p* = 0.0157, chi-squared test) based on the absence or presence of TAF in TCGA-CRC—especially CMS1/CMS4 (immune/mesenchymal) and PDS2 (inflammatory & stromal) subtypes are enriched in CRCs with TAF, which is really similar to SARIFA-positive CRCs. Boxplots in (**a**) depict median and interquartile ranges. x-Axis: presence (yes) vs absence (no) of TAF; y-Axis: mRNA expression z-scores relative to normal samples. Left: absence of TAF; Right: presence of TAF; blue: SARIFA-negative; red: SARIFA-positive. CD36: fatty-acid translocase, CMS consensus molecular subtype, CRC colorectal cancer, FABP4 fatty-acid binding protein, SARIFA Stroma AReactive Invasion Front Area, PDS pathway-derived subtype, TAF tumor adipose feature, TCGA The-Cancer-Genome-Atlas.

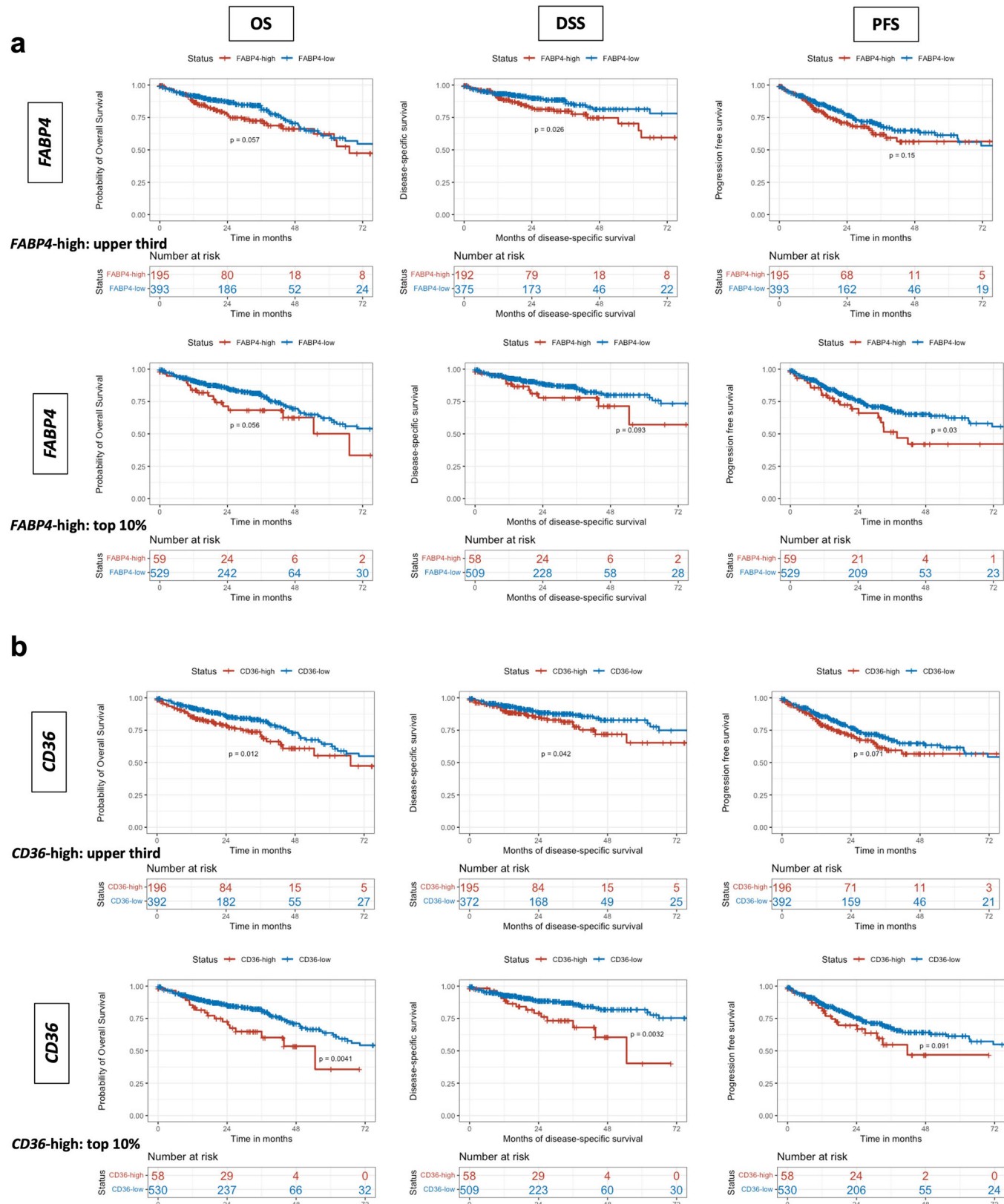

**Fig. 4 | Upregulation of *FABP4* and *CD36* gene expression is associated with poorer outcomes in TCGA-CRC.** CRCs with a high expression of *FABP4* or *CD36* (top 33.3% and top 10% with highest mRNA expression levels, respectively), both closely linked to lipid metabolism and SARIFA/TAF, are associated with poorer outcomes compared to low expressers with regards to most endpoints (OS, DSS, and PFS) in TCGA cohorts COAD and READ. **a** Kaplan–Meier curves stratified by *FABP4* expression, **b** Kaplan–Meier curves stratified by *CD36* expression. CD36 fatty acid translocase, COAD colonic adenocarcinoma, CRC colorectal cancer, DSS disease-specific survival, FABP4 fatty-acid binding protein 4, PFS progression-free survival, OS overall survival, READ rectal adenocarcinoma, SARIFA Stroma AReactive Invasion Front Area; TAF tumor adipose feature, TCGA The-Cancer-Genome-Atlas.

**Table 2 | TAF vs SARIFA: a comparison of adipose-associated histopathologic features/biomarkers**

| | Stroma AReactive Invasion Front Area SARIFA | Tumor Adipose Feature TAF |
|---|---|---|
| Origin of concept | Guided by histopathologic experience, proposed and validated as prognostic H&E-based histopathologic biomarker with an underlying characteristic tumor biology | machine-learning derived image feature, that was consecutively identified by pathologists and further validated as potential biomarker |
| Definition/ description | SARIFA[11–14]:= area at the tumor invasion front, in which at least one tumor gland or a group of at least five tumor cells are directly adjacent to adipocytes without interjacent inflammatory infiltrate or desmoplastic stromal reaction | TAF[19,26]:= small moderately-to-poor differentiated tumor cell clusters adjacent to a substantial component of adipose tissue (very similar feature described recently[27]: predominantly adipose and inflammatory cells with occasional tumor cells) |
| Location | Tumor invasion front | not applicable; identified on so-called tumor patches/tiles |
| Investigated entities | Gastric, Colorectal (pancreatic[38] & prostate[39] cancer) | Colorectal |
| kappa metric | in Colon Cancer: 0.87 and 0.77 in Gastric Cancer: 0.74 and 0.73 | in Colorectal Cancer: 0.69 (widespread vs others) |
| Findings | SARIFAs are independently prognostic in gastric and colon cancer, and associated with other high risk features SARIFAs are associated with an upregulation of lipid metabolism and an altered immune response | TAF are independently prognostic in a binary and also in a semi-quantitative way TAF are the first validation of a biomarker initially extracted from a machine learning model and then validated by human pathologists |
| *Conclusion* | SARIFA & TAF are both prognostic H&E image features/biomarkers that show similarities and overall are vivid and, as far as we know, first examples to really prove how histopathologic experience and machine learning can reinforce and benefit from each other. | |

Fig. 14)[41]. Another DL-based analysis linked a high proportion of adipocytes on H&E slides to poor prognosis and showed differences in the immune response between patients with high and low adipocytes[42]; however, in this study tumor-adipocyte interaction was not of relevance. Moreover, inflamed adipose tissue was also identified by training DL models to provide information regarding lymph node metastasis in early-stage CRC[43]. Still, not all DL models in CRC highlight tumor-adipocyte interaction as (most) important feature for prognosis, as Tsai et al. just recently showed that cancer-associated fibroblasts and stroma seemed to be of high importance for predictions in their model[44].

Even though SARIFA and TAF seem very similar on the first impression, we firmly believe that a thorough pathological review is necessary to link these concepts. By performing this review, we could indeed observe an overlap between SARIFA and TAF.

However, there were some subtle differences that may be relevant. For example, SARIFA is defined as a biomarker at the invasive front of the tumor, whereas TAF was initially described without such spatial context; nevertheless, the tumor-fat interface is most often located at the invasive margin of the tumor. Moreover, whereas SARIFA is defined by the direct contact between adipocytes and tumor cells without intervening stroma, TAF sometimes displays tumor cells that are separated from adipocytes by (desmoplastic) stroma (Fig. 2a upper panel). Additionally, sometimes in TAF, only mucin is located next to tumor cells, but no vital tumor, which is not in line with the SARIFA biomarker definition (Fig. 2a lower panel).

However, no statistically significant survival differences in SARIFA-negative patients with TAF vs without TAF could be seen (Fig. 2), indicating the crucial role of a direct tumor-adipocyte interaction for CRC patient prognosis. It is conceivable that the direct tumor-adipocyte interaction has tumor-promoting properties[12]; however, more research efforts and translational approaches deploying high-resolution techniques, such as spatial transcriptomics[45], are warranted to better understand if a direct tumor-adipocyte interaction is necessary for a tumor-promoting effect. From a diagnostic point of view, assessing direct tumor-adipocyte contact on H&E slides is more straightforward than taking adipocytes with close proximity to tumor cells into account (which may, for example, require cut-offs to establish how this proximity is defined). Nevertheless, the convenience of TAF lies in its automation, which does not require additional human effort. Based on this, further studies on real-world data need to be performed to ascertain whether TAF or SARIFA can be applied for clinical decision-making. Moreover, assessment of SARIFA/TAF in TCGA-CRC relied (mostly) only on one representative tumor slide, which is a limitation.

However, we have already shown that in most SARIFA-positive CRCs at least the majority of all resection slides shows the presence of SARIFA[13].

Additionally, we see a clear opportunity in the comparison provided here to enable pathologists and DL-based biomarkers to improve one another. For example, the DL-based concept of TAF could be further refined based on the growing understanding of SARIFA to focus on the features that are most biologically or prognostically relevant, perhaps leading to improved computational approaches for identifying SARIFA and prognostic tumor-adipocyte interaction.

In this study, we also developed a combined score (SARIFA/TAF: 1. double positive, 2. only TAF-present, 3. double negative), and could provide first insights that those CRCs with only TAF presence (close proximity of tumor cells to adipocytes but not direct interaction) may exhibit an intermediate phenotype regarding lipid metabolism (Supplementary Fig. 2). As described, those CRCs with only TAF did not show significantly reduced survival outcomes to those without TAF/SARIFA (double negative, Fig. 2). However, future and larger studies are necessary to better understand the prognostic and biologic relevance of tumor cells in close proximity to adipocytes without direct tumor-adipocyte contact.

Numerous basic and translational research studies have already highlighted the important role of lipid metabolism in tumor progression in general[8,9] and also with regard to CRC[7,46–48]. Moreover, targeting directly lipid metabolism, e.g., via FABP4 or CD36 inhibition, could open new therapeutic avenues in colorectal cancer[49–51]; especially as it is also conceivable to combine several lipid metabolism targeting drugs (FABP4/CD36 inhibitors) together with immunotherapeutic approaches or conventional chemotherapy in order to address several upregulated pathways at once in SARIFA-positive/TAF CRCs. Interestingly, in this study we found an association of increased *FABP4* and *CD36* mRNA expression with poor prognosis in CRC, further suggesting the tumor-promoting properties of lipid metabolism upregulation. Further, by using TCGA-CRC expression data we found an association between the presence of TAF and increased *FABP4* and *CD36* expression, just like the association with SARIFA-positivity in our previous works. Finally, the findings that TAF or SARIFA predict survival better than *FABP4* or *CD36* expression alone is in line with our findings that SARIFA is the morphologic correlate of a complex interplay of an upregulated lipid metabolism, immune dysregulation and a more stromal phenotype, that is based on the upregulation of several pathways[11,12,14,16]. We could validate this here by linking both H&E-based features SARIFA and TAF to transcriptional CMS1/4 and PDS2 subtypes[2,11,35,36]. Moreover, we provided further evidence for a biological overlap between SARIFA and TAF on a pathway level. Here, we

could show that SARIFA-positive CRCs seem to be characterized by an even more pronounced upregulation of lipid metabolism based on bulk RNA data. By linking H&E-based SARIFA/TAF classification to transcriptional CMS/PDS subtyping, we highlight the important role and potential of linking histopathology to underlying changes in gene expression.

To conclude, SARIFA and TAF independently validate each other as prognostic biomarkers, supporting tumor-adipocyte interaction as an important histopathologic biomarker that can be easily assessed on H&E slides and that could be implemented in the routine diagnostic workup to better stratify CRC patients. As far as we know, no histopathologic biomarker, that was initially discovered by DL models, could be implemented into routine pathologic practice. In a landscape where DL and histopathology unite, we present here a pivotal step forward in biomarker development by providing with SARIFA/TAF a paradigmatic example of a successful pathologist-AI interaction regarding biomarker discovery and validation. The slight differences between SARIFA and TAF might be explained by the fact that humans recognize very small features such as individual cells in contrast to the TAF tiles identified by DL reflect an integrated feature of several hundred cells. Nevertheless, the close similarity of TAF and SARIFA and the association with molecular markers suggest a high level of explainability and causability [52]. Our example demonstrates that the synergy of traditional histopathology and DL can lead to the identification of biomarkers that are clinically as well as biologically meaningful and that can potentially even lead to new therapeutic approaches. Whereas H&E histopathology is available for almost every cancer patient, there are large distributional disparities in access to molecular testing worldwide [53]. Therefore, we believe the development of novel H&E-based biomarkers in the interaction between pathologists and DL models is necessary to better stratify cancer patients, especially in settings where time- and cost-consuming extensive molecular testing is not available.

## Data availability

All source material on tumor adipose feature (TAF) as well as a similar image feature is fully available in the respective publications [19,26,27] and in the Supplementary Material and Supplementary Data included in this manuscript. To assess the overlap between SARIFA and TAF, several TAF patches, that were published with the corresponding articles (either in the main manuscript or as supplementary material), were investigated [19,26]. Clinical and molecular data, as well as image data of TCGA COAD (colonic adenocarcinoma) and READ (rectal adenocarcinoma), is publicly available at https://www.cbioportal.org/ [28–30]. For SARIFA-status in TCGA-CRC, please refer to our previous manuscript [11]. Source Data can be found in Supplementary Data 1–3. Furthermore, all data that has been generated during this study is included in this article and/or in its Supplementary Material/Data and/or available on reasonable request from the corresponding author (N.G.R.).

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

## Acknowledgements

JNK is supported by the German Federal Ministry of Health (DEEP LIVER, ZMVI1-2520DAT111), the German Cancer Aid (DECADE, 70115166), the German Federal Ministry of Education and Research (PEARL, 01KD2104C; CAMINO, 01EO2101; SWAG, 01KD2215A; TRANSFORM LIVER, 031L0312A; TANGERINE, 01KT2302 through ERA-NET Trans-can), the German Academic Exchange Service (SECAI, 57616814), the German Federal Joint Committee (TransplantKI, 01VSF21048) the European Union's Horizon Europe and innovation program (ODELIA, 101057091; GENIAL, 101096312) and the National Institute for Health and Care Research (NIHR, NIHR213331) Leeds Biomedical Research Centre. The views expressed are those of the author(s) and not necessarily those of the NHS, the NIHR, or the Department of Health and Social Care. NGR is supported by the Manfred-Stolte Foundation (gastrointestinal pathology research).

## Author contributions

N.G.R., B.G., and B.M. conceptualized this work. N.G.R. and B.G. performed the histopathologic review. N.G.R. and V.G. performed the bioinformatics analyses. V.G., D.S., B.G., D.S., E.W., V.L., M.P., H.M., K.Z.; H.S.M., J.N.K., and B.M. curated, interpreted, and/or analyzed the data. N.G.R. wrote the initial draft of the manuscript. All authors read and approved the final version of the manuscript.

## Funding

## Competing interests

J.N.K. declares consulting services for Owkin, France, DoMore Diagnostics, Norway, Panakeia, UK, Scailyte, Switzerland, Mindpeak, Germany, and Histofy, UK; furthermore, he holds shares in StratifAI GmbH, Germany, and has received honoraria for lectures by AstraZeneca, Bayer, Eisai, MSD, BMS, Roche, Pfizer and Fresenius; all of these are not related to this study. B.M. has received compensation for travel expenses and fees for advisory board activities by AstraZeneca, Boehringer Ingelheim, Merck, MSD, BMS, Bayer, and Novartis, not related to this study. K.Z. is founder and CEO of Zatloukal Innovations GmbH. D.F.S. and E.W. are current or past employees of Google LLC and own Alphabet stock.

## Additional information

[1]Pathology, Medical Faculty, University of Augsburg, Augsburg, Germany. [2]Bavarian Cancer Research Center (BZKF), Augsburg, Germany. [3]Else Kroener Fresenius Center for Digital Health, Technical University Dresden, Dresden, Germany. [4]Google Health, Google LLC, Palo Alto, CA, USA. [5]Department of Neurology, Ulm University, Ulm, Germany. [6]Department of Medicine and Surgery, Pathology, University of Milano-Bicocca, IRCCS (Scientific Institute for Research, Hospitalization and Healthcare) Fondazione San Gerardo dei Tintori, Monza, Italy. [7]Medical University of Graz, Diagnostic and Research Institute of Pathology, Graz, Austria. [8]Department of Visceral, Thoracic and Vascular Surgery, University Hospital Carl Gustav Carus Dresden, Dresden, Germany. [9]Pathology & Data Analytics, Leeds Institute of Medical Research at St James's, University of Leeds, Leeds, United Kingdom. [10]Department of Medicine I, University Hospital Dresden, Dresden, Germany. [11]Medical Oncology, National Center for Tumor Diseases (NCT), University Hospital Heidelberg, Heidelberg, Germany. ✉e-mail: nic.reitsam@uka-science.de; nic.reitsam@uk-augsburg.de

