## [Peer Review File · Communications Medicine]

Reviewers' comments:

Reviewer #1 (Remarks to the Author):

The article describes the overlap between two biomarker concepts: tumor adipose feature (TAF) and Stroma AReactive Invasion Front Areas (SAFIFA) which are both at a very early stage of development. The aim of the work is to investigate the overlap between human-observed SAFIFA and TAF. Both concepts have been published before and are not novel.

The study has a number of shortcomings that should be addressed:

SAFIFA: A case is classified as negative if there is “no direct tumor-adipocyte contact at any single site on the invasion front” (see Methods). I would like to understand how this criterion can be assessed with any degree of reliability on a single thin tissue section. Surely, this criterion depends heavily on where the given tissue section has been taken from, it is therefore subject to a sampling bias. If this criterion would be assessed on a 3D tissue section, I would follow the argument.

Comparison with CMS: First, the authors dismiss CMS by stating that the assay is too complex. At the same time they refer to imCMS [20], which predicts CMS subtypes on the basis of tissue morphology, but describe it as a ‘black box’ solution. It should be noted that CMS predictions can be generated at a tile level which does enhance the interpretability of this subtyping approach. I suspect that CMS3 captures metabolomic active tumors (see Figure 3 in [2])

The main article refers to patches without specifying at what size and resolution these are captured. It is also not clear how the different patches related to different cases. I did not understand how the information from different patches is being aggregated.

I could not figure out how concordance and discordance between the different reporting pathologists was either measured or reported. If I understand the text correctly, the histopathological assessment has been performed by one pathologist but “supervised” by others. It is not clear if this supervision refers to a rigorous consensus building or a mere observation. I would like to question this approach to biomarker development. Discordance among experts is certainly a concern in clinical reporting and needs to be addressed.

While it is true that the paper proposes a human-interpretable assessment of colorectal cancer histology I do not understand how this work leads to “new mechanistic insights”. In this context might also be useful to consider the work of Sandberg (DOI 10.1186/s12885-019-5462-2) and others who already comment on the role of FAB upregulated cells in the stroma.

Based on these comments I am uncertain how this paper contributes to the current state of the art and leads to any biomarker that can be translated into the clinic.

Reviewer #2 (Remarks to the Author):

In this paper, the authors investigated the overlaps between the tumor-adipose feature (TAF) and

the stroma areactive invasion front areas (SARIFA). These two concepts in colorectal cancer pathology were derived from deep learning analyses and human evaluations, respectively. The authors showed that TAF and SARIFA have significant overlap: based on the definition and empirical observations, almost all SARIFA image patches are also TAF patches. In patients with TAF but SARIFA-negative colorectal cancers, they observed poorer clinical outcomes. The authors also identified the associations between TAF and the expression levels of FABP4 (fatty-acid binding protein 4) and CD36 (fatty-acid translocase) using data from The Cancer Genome Atlas (TCGA). Below are my comments.

1. Because SARIFA is defined by visual evaluation of the microscopic images without resorting to quantitative tools, it would be interesting to quantify the level of inter-rater variability in evaluating samples from different sources.

2. The current analyses of FABP4 and CD36 expression employ two somewhat arbitrary cut-offs (the top-tertile or the 10% as the cut-off in the gene expression) to define patients with over-expressed genes. The authors could compare their current results with a standard analysis using fold changes to define the over-expression status of these genes.

3. The authors described a high-level overview of the reasons leading to discrepant TAF and SARIFA features in Table 1. More detailed analyses of the reasons for this discrepancy in specific datasets would further inform the reasons underlying the wide range of TAF/SARIFA non-overlap (0% to 100%; Table 1) observed in their studies. For example, what are the major reasons for TAF/SARIFA discrepancies in the L'Imperio et al. supplementary dataset (71% overlap)? Are they similar to the reasons for discrepancies in the Krogue et al. temporal validation set (0% overlap, which seems to indicate that all samples identified as TAF positive were labeled as SARIFA negative in this dataset)?

4. The authors could further investigate the expression levels of CD36 and FABP4 mRNA in colorectal cancer tissues with discordant TAF and SARIFA profiles. For example, in TAF-positive but SARIFA-negative colorectal cancer samples, do they exhibit high, intermediate, or low levels of CD36 and FABP4 expression? The authors could compare the gene expression levels of this subset of samples with those of (1) TAF-positive and SARIFA-positive and (2) TAF-negative and SARIFA-negative samples.

5. A few recent studies found pathology imaging features in the adipose tissues are associated with survival outcomes of colorectal cancer patients. For example, PMID: 35634419 and PMID: 37055393 showed that imaging features in the adipose tissue are indicative of overall survival outcomes. The authors could discuss the similarities and differences of these findings.

6. In addition to showing the overall correlation between the overall survival outcomes with FABP4 and CD36 gene expression levels, the authors could stratify their analyses by TAF and SARIFA features and investigate whether these gene expression profiles can independently predict survival outcomes in these subgroups.

7. Given the observation that TAF, SARIFA, and the expression levels of CD36 and FABP4 are correlated, the authors could elaborate on the potential synergistic (or antagonistic) effects of combining these predictors.

Dear editorial team of *communications medicine*, dear reviewers,

thank you for your time and providing us with constructive reviewer comments, to which we provide a point-by-point response below. Please note, that changes in the revised manuscript are highlighted in **yellow**.

In our revised manuscript, we provide now further analyses to better understand the biological overlap of SARIFA and TAF. Moreover, a second pathologist (BG) independently assessed the SARIFA/TAF overlap to validate our findings; a consensus diagnosis was reached if necessary.

For the sake of clarity, not all changes in the manuscript are repeated here in the rebuttal letter in their entirety.

Reviewer #1 (Remarks to the Author):

Rev1: The article describes the overlap between two biomarker concepts: tumor adipose feature (TAF) and Stroma AReactive Invasion Front Areas (SAFIFA) which are both at a very early stage of development. The aim of the work is to investigate the overlap between human-observed SAFIFA and TAF. Both concepts have been published before and are not novel.

The study has a number of shortcomings that should be addressed:

SAFIFA: A case is classified as negative if there is “no direct tumor-adipocyte contact at any single site on the invasion front” (see Methods). I would like to understand how this criterion can be assessed with any degree of reliability on a single thin tissue section. Surely, this criterion depends heavily on where the given tissue section has been taken from, it is therefore subject to a sampling bias. If this criterion would be assessed on a 3D tissue section, I would follow the argument.

Authors: The reviewer is correct that sampling bias and reducing tumor architecture from 3D in vivo into 2D on a histologic slide is always an important issue in histopathologic biomarker development (for example, refer to doi: 10.1111/his.12561.), and applies basically for the whole field of histopathology. In this context it must be noted that this is no ‘SARIFA -specific issue’, and is the same for example for tumor budding, which is a widely accepted biomarker: Unless you do not embed the whole tumor and do extensive serial sections of every FFPE tissue block, which is unrealistic for clinical practice, there is always the (low) chance of missing out some important morphology etc. However, we have already shown that in most SARIFA-positive CRCs the majority of all resection slides shows the presence of SARIFA (Martin et al. in *Cancers*). Therefore, assessing only one tissue section seems appropriate in the context of assessing SARIFA-status. Nevertheless, assessing only one representative tumor slide for SARIFA and TAF in TCGA-CRC is a limitation; we now included this in our discussion: *“Moreover, assessment of SARIFA/TAF in TCGA-CRC relied (mostly) only on one representative tumor slide, which is a limitation. However, we have already shown that in most SARIFA-positive CRCs at least the majority of all resection slides shows the presence of SARIFA¹⁰.”* We also rephrased our SARIFA-definition as the phrase “at any single site” was a bit misleading. We thank the reviewer for this valuable comment.

Rev1:Comparison with CMS: First, the authors dismiss CMS by stating that the assay is too complex. At the same time they refer to imCMS [20], which predicts CMS subtypes on the basis of tissue morphology, but describe it as a 'black box' solution. It should be noted that CMS predictions can be generated at a tile level which does enhance the interpretability of this subtyping approach.

Authors: We thank the reviewer for this important point. We definitely did not aim at dismissing CMS or imCMS, which are both extremely high-quality landmark studies in the field of CRC subtyping. CMS-subtyping based on RNA-expression analysis has not found its way into the clinic yet as RNA-expression analyses are only performed on few occasions in routine molecular pathology. However, the reviewer is completely right that novel approaches such as DL-based imCMS subtyping show a great potential as they can be applied on routine H&E stained tissue slides. The reviewer is also right that some DL-based approaches such as imCMS offer the great advantage of resolving molecular predictions on a tile level, offering interesting insights into tumor heterogeneity. We firmly believe that such promising DL-based approaches (as imCMS), in close correlation with histopathology, can lead to new findings, which our article aims to highlight using the example of SARIFA/TAF. We have therefore included the following section in the introduction: *“Even though these DL algorithms are very potent in predicting prognosis and molecular features, due to their partly ‘blackbox’ nature and the need for a computing-heavy infrastructure their implementation into the clinical diagnostic process is still very limited. However, by resolving molecular predictions, such as imCMS²⁰, on a spatial tile level, interesting insights into underlying tumor biology, including tumor heterogeneity, as well as increased interpretability, highlight the great potential of DL-models to gain biological insights. Beyond this, imCMS could help to identify patients who are most likely to benefit from neoadjuvant therapy²¹, further highlighting the potential of DL algorithms in CRC pathology.”*

Rev1: I suspect that CMS3 captures metabolomic active tumors (see Figure 3 in [2])

Authors: We understand the reviewer's idea. We have already investigated CMS-subtyping with regards to SARIFA-status in our previous paper and interestingly found an enrichment of CMS1/4 (immune/mesenchymal) within SARIFA-positive CRC, and no enrichment of CMS3 (metabolic) within SARIFA-positive CRCs. Even though CMS3 CRCs show indeed an upregulation of fatty-acid metabolism (such as SARIFA-positive CRCs), CMS3 CRCs show also an upregulation of many other metabolic pathways (see Figure 3 in Guinney et al.: Sugar, Glucose, Nitrogen, Glutamine, ... metabolism) and KRAS-activating mutations. SARIFA-positive CRCs are not only characterized by increased fatty-acid metabolism, but also dysregulated immunity and a stromal phenotype, leading to a clustering in CMS1/4. SARIFA-positivity (so far) seems not to be linked to a specific genetic background such as KRAS mutations.

To summarize, SARIFA-positivity is not associated with CMS3 subtype.

Based on this reviewer comment and to further validate our findings that SARIFA and TAF show an overlap, we also established now CMS-subtyping split by TAF presence. Moreover, we established novel pathway-derived subtyping (PDS) for SARIFA and TAF to prove that stromal & inflammatory changes (PDS2//CMS1/4) play an important role (besides upregulation of fatty-acid metabolism). We included these analyses in the revised manuscript. The results indicate that SARIFA/TAF are the morphologic correlate of an underlying tumor biology. This shows that H&E features (tumor-adipocyte interaction) identified by

pathologists/DL can give insights into the underlying tumor biology, and correlate with important biologic properties. Moreover, we performed pan-cancer immune subtyping (IS) based on the gene-expression signature, which did not show any SARIFA/TAF-dependent differences, indicating that site-specific transcriptomic classifiers may correlate better with histologic features identified by pathologists/DL methods. See in the manuscript (Figure 3, Figure S1) and in the text: *“Even though consensus molecular subtype (CMS) 3 is considered the metabolic CRC subtype with an upregulation of several metabolic pathways ((not only fatty-acid metabolism) and activating KRAS mutations², we have already demonstrated that SARIFA-positive CRCs show an enrichment for CMS1/4 (immune/mesenchymal) and not for CMS3⁸, indicating that SARIFA-positivity is not only associated with changes in fatty-acid metabolism^{8,9} but also dysregulated immunity^{11,12} and stromal changes. By investigating associations between CMS-calls and TAF (n=196 CRCs, TCGA-CRC), we also could, as expected, observe similar TAF-dependent differences in CMS-calls, with an enrichment of CMS1/4 (immune/mesenchymal, p=0.022, Figure 3C). By establishing novel pathway-derived subtypes (PDS)²⁸, we again observed statistically significant SARIFA/TAF-dependent differences (TAF: p=0.0157, SARIFA: p=0.004747), with an enrichment of PDS2 among SARIFA-positive CRCs/CRCs with TAF (Figure 3D, Figure S1A).*

Interestingly, compared to CMS/PDS Interestingly, pan-cancer immune subtyping (IS)²⁹ did not differ significantly based on SARIFA-status and TAF (each p>0.5), with most CRC cases in general being of ‘wound healing’ or ‘interferon-gamma dominant’ subtype (Figure S1B), potentially indicating that site-specific transcriptomic CRC classifiers (CMS & PDS) may better correlate with prognostically and biologically relevant histologic features (SARIFA/TAF).”

Rev1: The main article refers to patches without specifying at what size and resolution these are captured. It is also not clear how the different patches related to different cases. I did not understand how the information from different patches is being aggregated.

Authors: We thank the reviewer for this comment. We added the patch sizes in the revised manuscript:

- Patch size Wulczyn et al.: 256 × 256 pixels at 5× magnification (0.5 mm²)
- Patch size L’Imperio et al.: different resolution and size but scale bars are provided in the original publication for transparency
- Patch size Krogue et al.: 289 × 289 pixels obtained at 10X

Regarding aggregation of the patches and further information, we refer to the original publications. Patches do not necessarily belong to different patients; sometimes they are just most representative of their cluster, e.g. TAF.

Rev1: I could not figure out how concordance and discordance between the different reporting pathologists was either measured or reported. If I understand the text correctly, the histopathological assessment has been performed by one pathologist but "supervised" by others. It is not clear if this supervision refers to a rigorous consensus building or a mere observation. I would like to question this approach to biomarker development. Discordance among experts is certainly a concern in clinical reporting and needs to be addressed.

Authors: We understand the reviewer’s concerns as she/he is completely right that interobserver variability is an important issue in the development and clinical implementation of histologic biomarkers. However, we have already shown that assessment of SARIFA-status

in colorectal cancer shows a minimal interobserver variability (<10%, kappa-value up to 0.87, which is considered as very good interobserver agreement; Martin et al., *Cancers* 2021; <https://doi.org/10.3390/cancers13194880>). Identifying tumor glands/tumor cells next to adipocytes is an easy task for every pathologist, especially in CRC, and is not based on subjective assessments or estimates, like other established H&E biomarkers.

Nevertheless, the assessment of SARIFA/TAF overlap was now repeated by a histopathologic review of every patch by BG, another pathologist, who has published several articles on SARIFA assessment in gastric as well as colorectal cancers. For discrepant cases a consensus was reached. Kappa value was 0.87 indicating high interobserver agreement (see Table S1). All senior pathologists (BM, VI, DS, KZ), who developed SARIFA and TAF as biomarkers, supervised the assessment and plausibility. We added this in the manuscript.

Rev1: While it is true that the paper proposes a human-interpretable assessment of colorectal cancer histology I do not understand how this work leads to “new mechanistic insights”. In this context might also be useful to consider the work of Sandberg (DOI 10.1186/s12885-019-5462-2) and others who already comment on the role of FAB upregulated cells in the stroma.

Authors: We appreciate the reviewer’s perception that our manuscript proposes a human-interpretable assessment of CRC histology. We believe that linking SARIFA and TAF as independently developed biomarkers together does not only serve as an example of pathologist-AI interaction but specifically highlights the importance of tumor-adipocyte interaction in CRC, which has been neglected so far, and is likely a “*mechanistic insight*” based on our previous studies and several studies from basic research supporting the important role of tumor-adipocyte interaction in cancer progression.

There might be a misunderstanding regarding FAP/FABP4: The paper of Sandberg et al. focusses on FAP (fibroblast activating protein), which plays an important role with regards to cancer-associated fibroblasts, the tumor microenvironment and stromal reaction, but is not the same as FABP4 (fatty-acid binding protein 4), which we evaluated further. Investigating the role of FAP at SARIFAs could be an interesting project for the future but is beyond the scope of the current study.

Rev1: Based on these comments I am uncertain how this paper contributes to the current state of the art and leads to any biomarker that can be translated into the clinic.

Authors: We thank the reviewer for his/her constructive comments, which we addressed. We believe that the benefit of tumor-adipocyte interaction (SARIFA/TAF) is that it is a solely H&E-based histopathologic biomarker, which can be easily integrated into the clinical workflow without the need for further testing, no additional costs and high interobserver variability – in contrast to several other biomarkers (e.g., gene expression based biomarkers).

Reviewer #2 (Remarks to the Author):

Rev2: In this paper, the authors investigated the overlaps between the tumor-adipose feature (TAF) and the stroma areactive invasion front areas (SARIFA). These two concepts in colorectal cancer pathology were derived from deep learning analyses and human evaluations, respectively. The authors showed that TAF and SARIFA have significant overlap: based on the

definition and empirical observations, almost all SARIFA image patches are also TAF patches. In patients with TAF but SARIFA-negative colorectal cancers, they observed poorer clinical outcomes. The authors also identified the associations between TAF and the expression levels of FABP4 (fatty-acid binding protein 4) and CD36 (fatty-acid translocase) using data from The Cancer Genome Atlas (TCGA). Below are my comments.

Authors: We thank the reviewer for the constructive feedback, and ideas for further analyses, which we now included into the manuscript. Based on these insightful comments, we performed many additional analyses.

Rev1: 1. Because SARIFA is defined by visual evaluation of the microscopic images without resorting to quantitative tools, it would be interesting to quantify the level of inter-rater variability in evaluating samples from different sources.

Authors: The reviewer is correct that inter-observer variability is a crucial aspect in biomarker development. We have already shown that assessment of SARIFA-status in colorectal cancer shows a minimal interobserver variability (<10%, kappa-value up to 0.87, which is considered as very good interobserver agreement; Martin et al., *Cancers* 2021; <https://doi.org/10.3390/cancers13194880>). Identifying tumor glands next to adipocytes is a relatively easy task for every pathologist, especially in CRC. As assessment of SARIFA-status is based on conventional H&E histopathology used for routine diagnostics in daily pathologic practice, we had no problems in assessing SARIFA-status in different cohorts from different samples in our previous studies, even in different entities, namely gastric cancer, colorectal cancer, prostate cancer and pancreatic cancer; please refer to the following studies:

<https://doi.org/10.3390/cancers13194880>; <https://doi.org/10.3390/cancers15030994>;
<https://doi.org/10.1038/s41417-023-00695-y>; <https://doi.org/10.1002/path.5810>;
<https://doi.org/10.1007/s10120-023-01436-8>; <https://doi.org/10.1038/s41416-023-02515-4>; <https://doi.org/10.1186/s12885-023-11771-9>;
<https://doi.org/10.1101/2024.01.22.24301622>).

TAF/SARIFA overlap was now again assessed by another pathologist (BG); interobserver agreement was very good (kappa: 0.87). We added this in the revised manuscript (see also Table S1).

Rev2: 2. The current analyses of FABP4 and CD36 expression employ two somewhat arbitrary cut-offs (the top-tertile or the 10% as the cut-off in the gene expression) to define patients with over-expressed genes. The authors could compare their current results with a standard analysis using fold changes to define the over-expression status of these genes.

Authors: As we mention in our methods section, these analyses did not aim at finding an optimized cut-off for biomarker finding but rather to give a dose-dependent biological insight into the prognostic role of FABP4 and CD36. Optimized cut-offs do not necessarily provide these insights. We did not aim at establishing FABP4 or CD36 expression itself as prognostic markers. Nevertheless, we performed differential gene expression analysis via *DeSeq2* and now report log fold changes for FABP4 and CD36 regarding presence/absence of TAF, as the reviewer is completely right that log fold changes are here the more convenient and important

metric. Moreover, we used differential gene expression analysis to further characterize the biological overlap between SARIFA/TAF. Please *see point 4*.

Rev2: 3. The authors described a high-level overview of the reasons leading to discrepant TAF and SARIFA features in Table 1. More detailed analyses of the reasons for this discrepancy in specific datasets would further inform the reasons underlying the wide range of TAF/SARIFA non-overlap (0% to 100%; Table 1) observed in their studies. For example, what are the major reasons for TAF/SARIFA discrepancies in the L'Imperio et al. supplementary dataset (71% overlap)? Are they similar to the reasons for discrepancies in the Krogue et al. temporal validation set (0% overlap, which seems to indicate that all samples identified as TAF positive were labeled as SARIFA negative in this dataset)?

Authors: We agree with the reviewer that a more detailed analysis in this context is beneficial. We added this review in Table S2 for L'Imperio et al. and Krogue et al. In line with the Krogue et al. description of their TAF-like feature as “predominantly adipose and inflammatory cells with occasional tumor cells”, we found that some patches here did only contain adipose tissue and inflammatory cells without tumor cells (thus showing no SARIFA, and being rather in line with the Broeckmoeller et al. findings). For L'Imperio et al. as the pathologist-validation study the overlap between SARIFA/TAF is almost perfect, with only one case where there is a really small rim of stroma preventing direct tumor-adipocyte interaction.

Rev2: 4. The authors could further investigate the expression levels of CD36 and FABP4 mRNA in colorectal cancer tissues with discordant TAF and SARIFA profiles. For example, in TAF-positive but SARIFA-negative colorectal cancer samples, do they exhibit high, intermediate, or low levels of CD36 and FABP4 expression? The authors could compare the gene expression levels of this subset of samples with those of (1) TAF-positive and SARIFA-positive and (2) TAF-negative and SARIFA-negative samples.

Authors: We thank the reviewer for this important point, which triggered us to further analyse the biological overlap of SARIFA and TAF (please refer to our revised Figure 3 and Supplementary Figure S2). TAF-positive SARIFA-negative CRCs (only TAF) seem indeed to exhibit intermediate levels of CD36 and FABP4 expression (Figure S2). Our pathway analysis of overlapping and differing genes indicated that SARIFA-positive CRCs show a more pronounced upregulation compared to TAF-present (Figure 3). However, both have quite some similarities in their biological profile, which is logically as they also overlap quite a lot on a histologic level (Figure 3). We believe we now better grasp this biology in our revised manuscript and are very thankful for the reviewer's suggestion.

Rev2: 5. A few recent studies found pathology imaging features in the adipose tissues are associated with survival outcomes of colorectal cancer patients. For example, PMID: 35634419 and PMID: 37055393 showed that imaging features in the adipose tissue are indicative of overall survival outcomes. The authors could discuss the similarities and differences of these findings.

Authors: We thank the reviewer for those two interesting references. Tsai et al. rather highlight the importance of cancer-associated fibroblasts and stroma, which are also known to be important in CRC. We had already included the other paper in our initial manuscript but

now also discuss that here only adipocytes and not specifically tumor-adipocyte interaction is of relevance. Moreover, we included Jiang et al., which is another large end-to-end prognostication multicentric DL-based study, where tumor-infiltrated fat was highlighted by the DL model.

Rev2_ 6. In addition to showing the overall correlation between the overall survival outcomes with FABP4 and CD36 gene expression levels, the authors could stratify their analyses by TAF and SARIFA features and investigate whether these gene expression profiles can independently predict survival outcomes in these subgroups.

Authors: We thank the reviewer for this interesting comment. First, we could validate the prognostic value of both TAF and SARIFA in the subset (n=196), on which we now performed differential gene expression analysis (Figure S3). We then stratified this subset by top/low FABP4/CD36 expression based on the gene expression counts, and investigated if FABP4/CD36 can stratify within different subgroups (SARIFA-positive/negative, TAF-present/absent). Here, no significant survival differences could be observed. The results are paradigmatically visualized in Figure S4 and also in detail listed in Table S4.

Rev2: 7. Given the observation that TAF, SARIFA, and the expression levels of CD36 and FABP4 are correlated, the authors could elaborate on the potential synergistic (or antagonistic) effects of combining these predictors.

Authors: We thank the reviewer for this comment. We believe that our study shows that SARIFA could potentially be a refined TAF classifier (see discussion: *“For example, the DL-based concept of TAF could be further refined based on the growing understanding of SARIFA to focus on the features that are most biologically or prognostically relevant.”*). Our results show that SARIFA and TAF show quite an overlap (biologically and histologically).

Based on this comment, we now developed a combined score, and could show that those CRCs with only TAF may show an intermediate *FABP4* and *CD36* expression (Figure S2). Prognostically, we did not see statistically significantly reduced outcomes in those SARIFA-negative CRCs with only TAF compared to those without TAF (Figure 2). As *CD36* and *FABP4* are both upregulated in CRCs with TAF/SARIFA-positive CRCs, we believe that those two (*FABP4* and *CD36*) predictors are mainly encompassed within our TAF/SARIFA classification. See Figure S4 and Table S4 for prognostic value of further stratification.

Additionally, we believe that SARIFA-positivity and/or presence of TAF (and/or both on their own) could help to identify patients that could potentially benefit of drugs targeting lipid metabolism (such as *CD36/FABP4* inhibitors). One such drug (VT1021 – interfering with *CD36* and *CD47* via upregulation of TSP-1 expression) could induce at least a stable disease in a colorectal cancer case, published recently in *Communications Medicine* (<https://doi.org/10.1038/s43856-024-00433-x>). We included this reference and added the following sentence in the discussion: *“especially as it is also conceivable to combine several lipid metabolism targeting drugs (FABP4/CD36 inhibitors) together with immunotherapeutic approaches or conventional chemotherapy in order to address several upregulated pathways at once. Assessment of SARIFA/TAF status could serve here as appropriate biomarker.”* If SARIFA or TAF is more predictive for such potential treatment approaches in the future, needs

to be investigated in further studies (e.g. post-hoc analysis of clinical trials with regards to immunotherapy or chemotherapy).

REVIEWERS' COMMENTS:

Reviewer #2 (Remarks to the Author):

The authors have addressed the comments raised previously. Thank you.